# Pathobiological Relationship of Excessive Dietary Intake of Choline/L-Carnitine: A TMAO Precursor-Associated Aggravation in Heart Failure in Sarcopenic Patients

**DOI:** 10.3390/nu13103453

**Published:** 2021-09-29

**Authors:** May Nasser Bin-Jumah, Sadaf Jamal Gilani, Salman Hosawi, Fahad A. Al-Abbasi, Mustafa Zeyadi, Syed Sarim Imam, Sultan Alshehri, Mohammed M Ghoneim, Muhammad Shahid Nadeem, Imran Kazmi

**Affiliations:** 1Biology Department, College of Science, Princess Nourah Bint Abdulrahman University, Riyadh 11671, Saudi Arabia; mnbinjumah@pnu.edu.sa; 2Environment and Biomaterial Unit, Health Sciences Research Center, Princess Nourah bint Abdulrahman University, Riyadh 11671, Saudi Arabia; 3Department of Basic Health Sciences, Preparatory Year, Princess Nourah Bint Abdulrahman University, Riyadh 11671, Saudi Arabia; SJGlani@pnu.edu.sa; 4Department of Biochemistry, Faculty of Science, King Abdulaziz University, Jeddah 21589, Saudi Arabia; shosawi@kau.edu.sa (S.H.); fabbasi@kau.edu.sa (F.A.A.-A.); mzyadi@kau.edu.sa (M.Z.); mhalim@kau.edu.sa (M.S.N.); 5Department of Pharmaceutics, College of Pharmacy, King Saud University, Riyadh 11451, Saudi Arabia; simam@ksu.edu.sa (S.S.I.); salshehri1@ksu.edu.sa (S.A.); 6Department of Pharmacy Practice, College of Pharmacy, AlMaarefa University, Ad Diriyah 13713, Saudi Arabia; mghoneim@mcst.edu.sa

**Keywords:** sarcopenia, heart failure, trimethylamine-N-oxide, inflammatory mediators, choline, L-carnitine

## Abstract

The microecological environment of the gastrointestinal tract is altered if there is an imbalance between the gut microbiota phylases, resulting in a variety of diseases. Moreover, progressive age not only slows down physical activity but also reduces the fat metabolism pathway, which may lead to a reduction in the variety of bacterial strains and bacteroidetes’ abundance, promoting firmicutes and proteobacteria growth. As a result, dysbiosis reduces physiological adaptability, boosts inflammatory markers, generates ROS, and induces the destruction of free radical macromolecules, leading to sarcopenia in older patients. Research conducted at various levels indicates that the microbiota of the gut is involved in pathogenesis and can be considered as the causative agent of several cardiovascular diseases. Local and systematic inflammatory reactions are caused in patients with heart failure, as ischemia and edema are caused by splanchnic hypoperfusion and enable both bacterial metabolites and bacteria translocation to enter from an intestinal barrier, which is already weakened, to the blood circulation. Multiple diseases, such as HF, include healthy microbe-derived metabolites. These key findings demonstrate that the gut microbiota modulates the host’s metabolism, either specifically or indirectly, by generating multiple metabolites. Currently, the real procedures that are an analogy to the symptoms in cardiac pathologies, such as cardiac mass dysfunctions and modifications, are investigated at a minimum level in older patients. Thus, the purpose of this review is to summarize the existing knowledge about a particular diet, including trimethylamine, which usually seems to be effective for the improvement of cardiac and skeletal muscle, such as choline and L-carnitine, which may aggravate the HF process in sarcopenic patients.

## 1. Introduction

The human intestine microbiota is primarily comprised of four phyla: proteobacteria, firmicutes, actinobacteria, and bacteroidetes [1]. An imbalance between the gut microbiota phylases alters the microecological environment of a gastrointestinal tract, resulting in numerous diseases. The gut microbiota has many important functions in sustaining host fitness, including host feeding and energy harvesting, intestinal homeostasis, drug absorption and toxicity, immune system responsiveness, and pathogen defense. They can also produce microbial products such as bile acids, trimethylamine-N-oxide (TMAO), lipopolysaccharides (LPS), vitamin B complexes, vitamin K, uremic toxins, nitric oxide, fatty acids in the short-chain (SCFA), gut neurotransmitters, and hormones, which can modify host metabolism and influence both the health and diseases working in the body [2]. Moreover, progressive age not only slows down physical activity but also reduces the fat metabolism pathway, which may lead to a reduction in the variety of bacterial strains and bacteroidetes’ abundance, promoting *firmicutes* and *proteobacteria* growth. As a result, dysbiosis reduces physiological adaptability, boosts inflammatory markers, creates ROS, and induces the destruction of free radical macromolecules, leading to sarcopenia in older patients [3,4]. As aging became a global epidemic, decreased muscle mass in octogenarians (or older persons) impaired 5–13% of elderly people between 60 and 70 years old and has an incidence rate of up to 50% [5]. In a multi-continent sample, sarcopenia prevalence in the general population was between 12.6% and 17.5% [6].

Sarcopenia may be induced by heart failure via common pathogenetic pathways and mechanisms influenced by each other, such as physical activities, malnutrition, and hormonal changes. Prevalence levels are significantly greater in individuals with heart failure (HF), ranging between 19.5 and 47.3% [7].

Conversely, the development of heart failure may be favored by Sarcopenia via various mechanisms such as pathological ergoreflexes. It can be considered as a paradox that the association of sarcopenia is not visible with a sarcopenic cardiac muscle, while non-functional hypertrophy is displayed by cardiac muscles. In addition, cardiac hypertrophy can be considered as the normal mechanism of cardiac adaptation to the conditions of a rise in systemic demand. Cardiac dysfunctions can be caused by a hypertensive state in pregnancy and even in athletes via the heart’s physiological hypertrophy or via pathological hypertrophy, which can be triggered by various factors such as hemodynamic stress of irregular and prolonged nature, i.e., a hypertensive condition [4]. Cardiac cachexia has long been shown to be associated with decreased survival and this result can be considered independent of other prognostic variables such as low peak oxygen consumption, age, NYHA (New York Heritage Association) class, or LVEF (left ventricular ejection fraction) [8]. Additionally, research demonstrates a strong link between micronutrients such as Mg^2+^ and cardiovascular health, and highlights the potential pathophysiological pathways through which Mg^2+^ depletion may increase the development, progression, and maintenance of CVD. Indeed, hypomagnesemia has a detrimental effect on cardiovascular health, as it is linked with an increased prevalence of hypertension, type 2 diabetes, dyslipidemia, atherosclerosis, arrhythmias, and coronary artery disease [9], all of which are common in sarcopenia [10].

Enhanced muscle reflex has a significant link with peripheral muscle wastage and, additionally, the overactivity of muscle reflex can be considered consistent with the idea that the development of a syndrome is linked to the muscle’s peripheral maladaptive changes. There are some important factors, such as progressive age, associated with sarcopenia and the change in gut microbiota diversity. Dysbiosis can also be considered as an independent cardiovascular risk factor and as responsible for heart failure in elderly people. Minimal investigations have been conducted in elderly patients regarding the actual mechanisms, such as concerning cardiac mass alteration and dysfunction, which are considered equivalent with cardiovascular diseases. They can be concluded as the downward spiral of dysregulation regarding exercise of the skeletal muscle, which is suggested by the hypothesis of muscle and can be correlated with certain vicious cycles in heart failure in which, initially, there are adaptive physiological responses that are gradually converted into maladaptive responses [11]. Thus, the purpose of this review is to summarize the existing knowledge about a particular diet including trimethylamine, which usually seems to be effective for the improvement of cardiac and skeletal muscle, such as choline and L-carnitine, which may aggravate the HF process in sarcopenic patients.

## 2. Consideration of the Sources for the Review of Literature

Certain databases such as Medline, Mendeley, Google Scholar, Public Library of Science, PubMed, ScienceDirect, and Springer Link were considered and searched through for the literature review, searching for studies that were potentially relevant and in which certain keywords were used both alone and in conjunction. Certain keywords that were significant and were used for the search of literature were ‘Sarcopenia’, ‘Epidemiology of sarcopenia’, ‘Mechanism of sarcopenia mediated heart failure’, ‘Involvement of dysbiosis in sarcopenia’, ‘Pathogenesis of heart failure’, ‘Reactive oxygen species-mediated mitochondrial dysfunction, ‘Relationship of choline and L-carnitine for muscle function improvement’ or ‘Role of TMA and TMAO in heart failure, in combination with ‘heart failure and dysbiosis’, ‘Immunogenic profile in sarcopenia and heart failure’, and ‘ergoreflx mechanism in sarcopenia associated heart failure’. In this review, only papers in English were considered. The reference list of the papers found were also screened for related articles not detected by the initial search strategy.

## 3. Clinical Characteristics of Sarcopenia in Association with Gut Microbiota Diversity

Sarcopenia can be referred to as the gradual loss in mass of skeletal muscle, the loss of its strength, and the loss of functions performed, and it is now considered as the major factor of negative effects of health in the later period of life [12]. In fact, the high pervasiveness of chronic health conditions can be correlated with old age (e.g., inflammatory irritable bowel syndrome, celiac disease, autoimmune disease, colitis, diabetes, cancer, cardiovascular disease, neurodegeneration, and so on), which lead in turn to many negative health events (e.g., illness, loss of freedom, institutionalization, underprivileged quality of life, and mortality) [13,14,15].

The authors established a link between health status, diet, and microbiota. To be more precise, the composition of the microbial population was predominantly influenced by fruit, meat, and vegetable intake. Additionally, a higher proportion of two dominant phyla, namely Firmicutes (64%) and Bacteroidetes (23%), comprise up to 90% of the overall gut microbiota in older people who are living in long-term care facilities [16,17,18]). It has been identified that the level of Staphylococcus spp. and Lactobacillus Reuters, both of which are from phylum firmicutes, is high in obese people. A positive correlation has been established between plasma > C-reactive protein (CRP) and plasma [19,20]. Moreover, older people are primarily affected by a rise in Escherichia (phylum of proteobacteria) abundance [16]. However, it is understood that an increase in gram-negative bacteria such as proteobacteria in their relative abundance is one of the most significant harmful age-changes for the human intestinal microbiota composition, as lipopolysaccharides are secreted by these gram-negative bacteria, through which inflammation can be induced in the intestines [21]. Advancing age can also be characterized by a gastrointestinal microbiota’s dysbiosis, which promotes the circulation passage of endotoxin and other microbial products or metabolites via the increased permeability of the intestine [22], thereby highlighting the influential role of gut dysbiosis for deficits in muscle functions associated with age. Sarcopenic patients have increased serum c-reactive protein (CRP) levels, while trials with other inflammatory mediators such as interleukin 6 have not shown consistent results [23].

In addition, the maintenance of sarcopenia is supported by the insufficient nutritional system and aged immune system, which play key roles in stimulating the activation of chronic inflammation [24,25]. In cachexia and sarcopenia, however, mitochondrial and systemic inflammation plays a central role. The proinflammatory role of cytokines (e.g., IL6, IL1β, TNF-α, and TNF-style weak apoptosis inducer (TWEAK)) has previously been reported in inducting muscle catabolism [26] (Figure 1).

## 4. Dietary Intake Choline and L-Carnitine-Mediated Aggravation of CVD

Choline/L-carnitine was investigated as an ergogenic aid for improving the training ability of a stable athletic population due to its pivotal role in the oxidation of fatty acids and energy metabolism. Beneficial impacts on acute physical performance, such as increased power production and increased intake of maximum oxygen, were observed in earlier research studies and further studies show the beneficial influence of L-carnitine as a dietary supplementation in the post-exercise recovery process. L-carnitine has been shown to alleviate the injury of muscles and condenses’ cellular damage markers, and muscle soreness attenuation is accompanied by free radical formation [27].

In 2013, researchers first demonstrated that a molecular metabolite, namely trimethylamine-N-oxide (TMAO) isolated from the microbiota of the gut, predicted that 4007 healthy cardiac patients will be enduring elective coronary angiography with an excepted increased risk of cardiovascular accidents [28]. TMAO is produced by microbiota via the ingestion of meat products containing nutritional precursors of trimethylamine, such as phosphatidylcholine, glycerophosphocholine, trimethylglycine, betaine, γ-butyrobetaine, crotonobetaine, choline, and L-carnitine [28,29,30,31]. Specific intestinal microbial enzymes convert these precursors into trimethylamine and to date, they have identified four different types of microbial enzyme systems including choline-TMA lyase (cutC/D) [32], carnitine monooxygenase (cntA/B) [33], betaine reductase [34], and TMAO reductase [35]. Recently, it has also been demonstrated that elevated L-carnitine, choline, and phosphatidylcholine amounts reflect multiple cardiovascular hazards such as myocardial infarction, hypertension, atherosclerosis, and diabetes [36,37,38,39,40,41,42,43].

Change in the microbiota composition of the gut caused by sarcopenia and heart failure can alter the circulating levels of TMAO. Moreover, it has been identified that hypertension patients experience an alteration in intestinal microbiota diversity. Experiments conducted on rats who were treated with angiotensin II revealed that intestinal biota species were less diverse and when compared to regulated rats, the Firmicutes to Bacteroidetes ratio was increased [44,45]. Moreover, heart failure was considered a chronic systemic inflammatory disorder, which indicates a substantial rise in pro-inflammatory cytokines of plasma; although its origin is still unclear, this unresolved inflammation can be considered as one of the key components of cardiovascular diseases [46,47]. Several occurring signs indicate that the microbiota of the gut produces bioactive metabolites including bile acids, short chains of fatty acids, and TMAO, and might have systemic effects on the host [48]. Microbiotas and their metabolites affect intestinal health and other physiological processes, especially within the circulatory system. Under normal circumstances, most can be considered as healthy and safe bacterial metabolites, but due to the involvement of heart-failure-related cardiovascular pathologic processes, there is a risk of disruption in the balance of the microbiota of the gut as well as a risk of a rise in the level of harmful metabolites; generally, it was shown in studies that TMAO was found to be related with the prognosis of at-risk heart-failure patients. Moreover, Firmicutes, including Enterococcus, Proteobacteria, Anaerococcus, Streptococcus, and Desulfitobacterium including Actinobacteria, Clostridium, Citrobacter, Dseulfovibrio Enterobacter, Escherichia, Proteus, Pseudomonas, and Klebsiella, have been linked with the production of the primary component of TMAO, i.e., TMA [49].

One research study found that eight Firmicute and Proteobacteria species have absorbed more than 60% of the production of choline of TMA, including Escherichia fergusonii, Clostridium asparagiforme, C. hathawayi, C. sporogenes, Edwardsiella tarda Anaerococcus hydrogenalis, Proteus penneri, and Providencia rettgeri [50]. Akkermansia, Prevotella, and Sporobacter are some other gut microbiota that are associated with the higher production of TMAO [51], and atherosclerotic CAD is associated with Ruminococcus gnavus [52]. The growth of CAD may be predicted via different metabolites such as betaine, choline, and TMA. It can be explained, for instance, by considering that TMAO-producing microbes can be reduced by blocking or inhibiting specific microbial metabolic pathways via utilizing pharmacological intervention and probiotics [53]. Furthermore, the increased level of *Ruminococcus* is due to the high fat and high protein diet [54], and additionally, downregulation of Treg cells is led by TLR4 activation, which is associated with inflammatory responses such as CD4, Pro-inflammatory cytokines, and Th1 and T cells [55,56]. Thus, we explore, from top to bottom, all of the contributing factors associated with CVD.

## 5. Pathobiological Interactions in Heart Failure Involving TMAO

Mechanisms of heart failure pathophysiological pathways are quite intricate and include inflammatory reaction, hemodynamics irregularity, cardiac remodeling, neuroendocrine system stimulation, etc. Traditionally, the key causes of heart failure are supposed to be the activation of the pathways of the neuroendocrine system, which include the natriuretic peptide system, renin-angiotensin-aldosterone cascade, and sympathetic nervous system, which lead to a pathologic myocardial remodeling process series including apoptosis, extracellular matrix deposition, myocardial hypertrophy, and resultant fibrosis [57,58]. Hence, neuroendocrine inhibition is the main basis of the strategies of current treatments [59]. Mechanisms driving the development and progression of heart failure are, however, still under consideration. In the conversion of dietary choline into the intermediate trimethylamine (TMA), a requisite role is played by microbiota of the gut and TMAO is formed by the subsequent oxidization of TMA after it enters into the circulatory system by the flavin-containing monooxygenase (FMO) enzyme, which is encoded by the FMO gene present in the kidney, liver, and in many other tissues [60,61]. There is an increase in the permeability of the intestinal barrier via two mechanisms in the condition of heart failure, in which during the initial stage, a decreased inflow of blood to the intestinal endothelium is observed, and via the ischemia of the wall of the intestine, there is an increase in the permeability of the intestinal epithelial barrier [62]. Due to the intestinal wall’s congestion and swelling in the advanced stages of heart failure, there is an increase in the permeability of the intestine. Additionally, in the patients identified with chronic heart failure, higher levels of enteropathogenic candida, such as Campylobacter, Shigella, and Salmonella, were observed [63]. This process is directly linked with microbial and microbial metabolite translocation [64,65]. Recent research evidence indicates that chronic inflammation can be caused by both an increase in the permeability and an increase in the disordered microbiota of the intestine, further leading to impaired cardiac function [62,66]. In addition, studies have shown that there are severe clinical symptoms and worse survival rates associated with patients with heart failure, which are due to the elevated serum levels of multiple cytokines, such as IL-1, IL-6, and the TNF [67,68,69]. This is consistent with findings that both heart failure and sarcopenic patients have an elevated proportion of these bacterial strains of the intestine, indicating shifts in intestinal microbiota, which may influence levels of TMAO by controlling intestinal TMA synthesis. TMAO has recently become a major mediator showing that the microbiota of the gut has a close relationship with several CVDs. Subsequent preclinical experiments explored the evidence concerning that the heart is directly affected by the TMAO, inducing endothelial cell and vascular inflammation, fibrosis and myocardial hypertrophy, and heart mitochondrial dysfunction, thus aggravating the heart-failure process [70,71,72]. In addition, the association of TMAO is established with both the C-reactive protein (CRP) and with endothelial dysfunction in evaluating the increased permeability of the gut, and is closely related to increased LPS endotoxin serum levels [49], leading to the release of calcium and the hyperreactivity of the platelets [73], contributing to the aggravation of heart failure. The several key pathophysiological pathways of TMAO include the following: explicitly and implicitly contributing in heart failure, including through the pathological LV dilation of the mouse-fed TMAO or choline-demonstrated decreased LVEF, and enhanced circulatory BNP volumes, myocardial fibrosis, and lung oedema [31]; TMAO-encouraged myocardial hypertrophy and fibrosis through Smad3 signals [71]; cardiac remodeling attenuated through 3,3-dimethyl-1-butanol via the reduction in the volume of plasma TMAO, which modifies the signals of TGF-β1/Smad3 and p65 NF-kB [74]; TMAO-promoted activated leukocyte recruitment into endothelial cells and induced inflammatory gene expression via the activation of NF-kB signaling [75]; TMAO significantly affected the contractile nature of cardiomyocyte and intracellular calcium-handling in the negative direction [76]; Pyruvates and fatty acid oxidation in cardiac mitochondria is influenced by TMAO [70]; and, last but not least, TMAO stimulated vascular inflammation by triggering the inflammatory NLRP3 induced by inhibiting SIRT3-SOD2–mitochondrial ROS signaling pathway [77]. Moreover, the function of TMAO, as first assessed by Suzuki et al. [78] in acute HF (AHF), was found to be a predicting marker for mortality and mortality/heart failure within a year (Table 1) [79].

Additionally, an independent cohort of ambulatory individuals with persistent systolic HF supports our results and provides new insights on the link between the three phosphatidylcholine metabolic isomers, namely TMAO, choline, and betaine, considering echocardiographic determinants and the associations between both renal and inflammatory biomarkers. Numerous noteworthy discoveries have been made. To begin, we found that TMAO had a superior predictive value to choline and betaine in patients with chronic systolic heart failure, regardless of the cardio-renal parameters. Second, rather than LV systolic dysfunction, we found associations between all three metabolites and LV diastolic dysfunction. Thirdly, the very low correlations between TMAO, choline, and betaine in many well-characterized inflammatory biomarkers and in their distinct associations with endothelial dysfunction indicators indicated the existence of a separate pathophysiological mechanism. Notably, the increased TMAO levels seen in individuals with renal insufficiency or diabetes mellitus suggest an underlying metabolic deficiency associated with those disease states rather than a systemic inflammatory response. Nonetheless, the relationship between increased TMAO and both HF severity and adverse outcomes, irrespective of other cardio-renal indices, argues for a possible harmful molecular link between the gut microbiota pathway that generates TMAO and the development and/or progression of HF. Notably, this is a cohort of ambulatory stable heart failure patients with left ventricular systolic dysfunction and with an annualized mortality of 7.1% (considering transplantation as the equivalent of death), which is not dissimilar to that seen in published clinical trials. Taken together, our results validate the clinical relevance of TMAO levels in heart failure and indicate that further research is needed to elucidate the association’s molecular underpinnings. However, after tuning for the parameters of renal function, the capacity of the TMAO to independently forecast is lost, likely due to the substantial correlations between the parameters of renal function (approximate glomerular filtration rate and urea) and TMAO. These findings indicate that a higher degree of “backward failure” (congestion associated with scarring or ischemia) rather than “forward failure” (or reduced perfusion) may be linked with the main metabolic deficiency underlying the observed correlations. Consistent with this, correlations between choline and renal function indices were seen for both choline and TMAO, although the link between TMAO and adverse outcomes in individuals persisted even after adjusting for renal function. The purpose of this study was to investigate the connection between (1) the intestinal microbiota-dependent analyte TMAO and its dietary precursors, namely and choline and betaine, and (2) echocardiographic indicators in sarcopenic patients with chronic systolic heart failure [80,81] (Figure 2).

## 6. Conclusions

Sarcopenia is common in cases of heart failure, leading to inadequate disease prognosis. While the pathophysiology of muscle wastage is quite complicated in heart failure, multiple pathogenetic mechanisms tend to be shared by sarcopenia and heart failure, and they can benefit from strategies of standard treatment focused on a nutritional, physical, and pharmacological approach. In recent years, several studies have identified a clear correlation between CVDs and the microbiota of the gut. We already know that TMAO, a gut microbiota metabolite, may have fresh perspectives and insights regarding how heart failure is supported by the microbiota of the gut. These findings provide a good opportunity for controlling heart failure via addressing the microbiota of the gut, including through the use of updated probiotics, prebiotics, dietary therapy, and FMT. Moreover, emerging research from different groups and clinical findings reveal the association between the dysfunction of the microbiota of the gut, the TMAO circulation, and the susceptibility of heart failure, indicating a fresh and desirable therapeutic target for HF treatment. Furthermore, excessive intake of a diet such as choline or L-carnitine, which contain intermediate precursor TMA for TMAO, should be carefully used in elderly people who have dysbiosis with muscle disorders. Future research studies are warranted.

## Figures and Tables

**Figure 1 nutrients-13-03453-f001:**
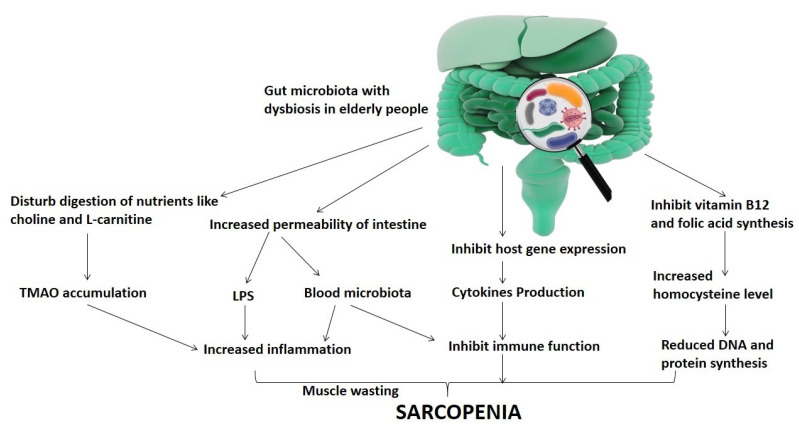
Representation of the relationship of gut microbiota dysbiosis-mediated sarcopenia in elderly people.

**Figure 2 nutrients-13-03453-f002:**
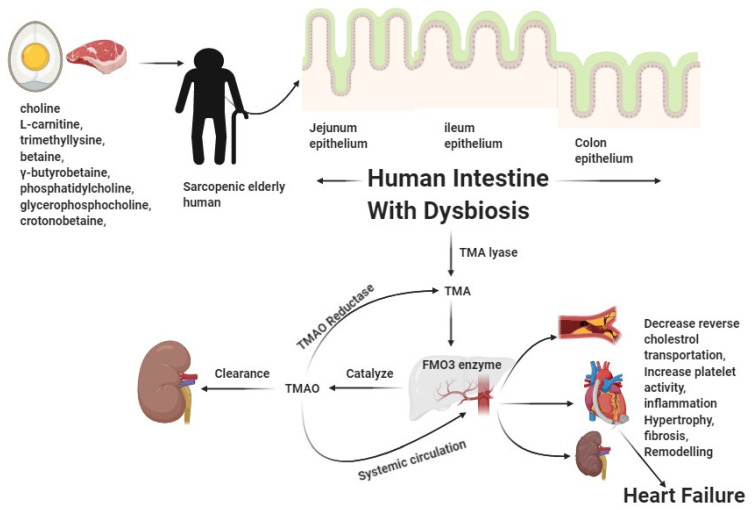
Representation demonstrating the pathobiological relationship of the excessive intake of choline/L-carnitine-containing diet-associated TMAO accumulation, resulting in the heart failure of sarcopenic patients.

**Table 1 nutrients-13-03453-t001:** TMA metabolism-targeting therapeutic methods.

Therapy	Alteration in Biotransformation TMA	Implications
Inhibition of the FMO3 enzyme	Prevents oxidation of TMA to TMAO	Trimethylaminuria is caused by an accumulation of TMA and is characterized by a fishy odor. It may also cause inflammation. Additionally, FMO3 metabolizes a wide variety of other compounds.
Resveratrol	Modifies the makeup of the gut microbiota. Reduces the formation of TMA and TMAO	Increases *Lactobacillus* and *Bifidobacterium.* When antibiotics are taken, no adverse effects occur. Observed in mice studies.
Enalapril	Increases TMAO excretion in the urine	Mechanism unknown. Rat studies were conducted. It does not affect TMA synthesis or the makeup of the gut flora.
Prebiotics	Induces a beneficial effect on the makeup of the gut bacteria to reduce TMA production in the intestine	In humans, the consequences are unknown. Numerous variables affect the makeup of the gut microbiota.
Probiotics (I): Methanogenic bacteria	Reduces TMA and TMAO levels	Human safety and engraftment are unknown.
Probiotics (II): Bacteria incapable of converting precursors to TMA	Reduces the production of TMA in the gut	Mice show beneficial benefits. However, the consequences on people remain unknown.
Meldonium	Reduces the production of TMAO from L-carnitine (GBB conversion to L-carnitine is inhibited)	TMAO production from choline cannot be reduced. It may result in a rise in the urine excretion of TMAO in people.
Oral non-absorbent binders	Eliminates TMAO or any of its precursors from the gut	A speculative approach. There has not yet been found a chemical capable of removing TMAO specifically.

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
