# Peer review of "Pathobiological Relationship of Excessive Dietary Intake of Choline/L-Carnitine: A TMAO Precursor-Associated Aggravation in Heart Failure in Sarcopenic Patients"

_nutrients, 2021, doi:10.3390/nu13103453_

Round 1

Reviewer 1 Report

The manuscript from Sadaf Jamal Gilani describes the contribution of TMAO formation from gut microbiota in the development of heart failure and the potential implication of choline/carnitine support as a disease-modifier component. The field is of potential interest. However, very consistent changes are required before the manuscript may be considered suitable of publication.

1) Very extensive editing of English language and style required

2) The mechanisms of TMAO generation and their implications in the cross talk between sarcopenia and HF are poorly discussed

3) Evidences in pre-clinical and clinical settings of the contribution of choline/carnitine supplementation are lacking. Tables summarising data in the literature could help the reader to better assess their role in HF and sarcopenia.

Author Response

English language and style

(x) Extensive editing of English language and style required
( ) Moderate English changes required
( ) English language and style are fine/minor spell check required
( ) I don't feel qualified to judge about the English language and style

Response: Revised language extensively

Comments and Suggestions for Authors

The manuscript from Sadaf Jamal Gilani describes the contribution of TMAO formation from gut microbiota in the development of heart failure and the potential implication of choline/carnitine support as a disease-modifier component. The field is of potential interest. However, very consistent changes are required before the manuscript may be considered suitable of publication.

  • Very extensive editing of English language and style required

Reply: I have revised a whole manuscript and improved an English language.

2) The mechanisms of TMAO generation and their implications in the cross talk between sarcopenia and HF are poorly discussed

Reply: Dear reviewer, as per your instructions.  I have the “mechanisms of TMAO generation and their implications in the cross talk between sarcopenia and HF” in last paragraph of manuscript.

3) Evidences in pre-clinical and clinical settings of the contribution of choline/carnitine supplementation are lacking. Tables summarising data in the literature could help the reader to better assess their role in HF and sarcopenia.

Reply: Dear reviewer, as per your instructions I have included a table of Evidences in pre-clinical and clinical settings of the contribution of choline/carnitine supplementation.

Reviewer 2 Report

The review is potentially interesting and it focuses on a rarely discussed and often confusing argument. Some aspects should be improved:

1) an extensive english editing is required both from grammatical and syntactic point of view.

2) some sentences are confusing and and need to be reorganized.

3) please discuss the role of oligoelements in the prevention of CVD such as magnesium (please see  2019 May 2;2019:4874921.doi: 10.1155/2019/4874921.)

4) the discussion regarding heart failure syndrome should be improved

Author Response

English language and style

(x) Extensive editing of English language and style required
( ) Moderate English changes required
( ) English language and style are fine/minor spell check required
( ) I don't feel qualified to judge about the English language and style

Response: Language revised extensively.

Comments and Suggestions for Authors

The review is potentially interesting and it focuses on a rarely discussed and often confusing argument. Some aspects should be improved:

  • an extensive english editing is required both from grammatical and syntactic point of view.

Reply: I have revised a whole manuscript and corrected a grammatical mistakes.

2) some sentences are confusing and and need to be reorganized.

Reply: I have corrected a sentences.

3) please discuss the role of oligoelements in the prevention of CVD such as magnesium (please see  2019 May 2;2019:4874921.doi: 10.1155/2019/4874921.)

Reply: I have included a role of oligoelements in the prevention of CVD such as magnesium in introduction section

4) the discussion regarding heart failure syndrome should be improved

Reply: I have extended a discussion section as per your instructions.

Round 2

Reviewer 1 Report

The manuscript has been improved and now is suitable of publication